# DNA methylation–independent long-term epigenetic silencing with dCRISPR/Cas9 fusion proteins

Li Ding[1], Lukas Theo Schmitt[1], Melanie Brux[1,2], Duran Sürün[1], Martina Augsburg[1], Felix Lansing[1] ●, Jovan Mircetic[1,3], Mirko Theis[1,2], Frank Buchholz[1,2,4] ●

The programmable CRISPR/Cas9 DNA nuclease is a versatile genome editing tool, but it requires the host cell DNA repair machinery to alter genomic sequences. This fact leads to unpredictable changes of the genome at the cut sites. Genome editing tools that can alter the genome without causing DNA double-strand breaks are therefore in high demand. Here, we show that expression of promoter-associated short guide (sg) RNAs together with dead Cas9 (dCas9) fused to a Krüppel-associated box domains (KRABd) in combination with the transcription repression domain of methyl CpG–binding protein 2 (MeCP2) can lead to persistent gene silencing in mouse embryonic stem cells and in human embryonic kidney (HEK) 293 cells. Surprisingly, this effect is achievable and even enhanced in DNA (cytosine-5)-methyltransferase 3A and 3B (Dnmt3A$^{-/-}$, Dnmt3b$^{-/-}$) depleted cells. Our results suggest that dCas9-KRABd-MeCP2 fusions are useful for long-term epigenetic gene silencing with utility in cell biology and potentially in therapeutical settings.

## Introduction

The CRISPR/Cas9 system has emerged as a powerful and versatile genome manipulation tool. Initially, Cas9 has been programmed with short guide RNAs (sgRNAs) to introduce DNA double strand breaks at specific genomic target sites to generate deletions or insertions through host cell DNA repair systems (Knott & Doudna, 2018). However, soon after the original use as a DNA nuclease, scientists discovered that the CRISPR/Cas9 system represents a versatile platform for additional advanced genome alterations (Hashemi, 2018). Examples are the development of base and prime editing approaches that allow for the precise and predictable modification of DNA without the introduction of DNA double strand breaks (Anzalone et al, 2020). Base editors induce transition substitutions of targeted bases by fusing nickase Cas9 (nCas9) or catalytically inactive Cas9 (dCas9) with a nucleobase deaminase enzyme. Two main classes of base editors include cytosine base editors and adenine base editors that have been used to mediate all four transition mutations (Anzalone et al, 2020). The recently developed prime editors offer more flexibility and can be used to convert any nucleotide or introduce desired indels. Prime editors use an engineered reverse transcriptase (RT) fused to a nCas9, and a prime editing guide RNA (pegRNA). Mutations are introduced via DNA synthesis by RT using pegRNA with desired sequence changes as a template and the nicked DNA strand annealed to the pegRNA as a primer (Anzalone et al, 2019; Sürün et al, 2020).

Other examples for CRISPR/Cas9 as a platform for genome alterations are CRISPRa (CRISPR activation) and CRISPRi (CRISPR interference). Here, dCas9 is fused to transcriptional enhancers or repressors. In combination with a specific sgRNA, the fusion-proteins are recruited to promoters/enhancers of genes to modulate their transcription. For CRISPRa, transcription activation frequently uses multimers of the VP16 minimal activation domain derived from Herpes simplex virus (e.g., dCas9-VP64), which can substantially increase transcription when guided to the promoter region of a gene (Chavez et al, 2015). In contrast, the Krüppel-associated box domain (KRAB) of the transcription factor KOX1 has been widely adopted for CRISPRi experiments. The fusion of this KRAB domain to dCas9 and subsequent targeting to a promoter or enhancer mediates reversible transcriptional repression (Groner et al, 2010). However, the KRAB domain from KOX1 is not a particularly strong repressor, and the dCas9–KRAB system suffers from inefficient knockdown (Evers et al, 2016).

Fusion of several transcriptional regulators to dCas9 in tandem can synergize to increase their repressive activities (Chavez et al, 2016). To improve the performance of gene repression, Yeo and colleagues designed a bipartite repressor cassette by fusing the KOX1 KRAB domain with the transcription repression domain of MeCP2 and investigated its silencing efficiency when targeted to

[1]Medical Systems Biology, Medical Faculty and University Hospital Carl Gustav Carus, TU Dresden, Dresden, Germany [2]National Center for Tumor Diseases (NCT/UCC) Dresden, German Cancer Research Center (DKFZ), University Hospital Carl Gustav Carus, Technische Universität Dresden, Helmholtz-Zentrum Dresden-Rossendorf (HZDR), Dresden, Germany [3]Mildred Scheel Early Career Center, National Center for Tumor Diseases Dresden (NCT/UCC), Medical Faculty and University Hospital Carl Gustav Carus, TU Dresden, Dresden, Germany [4]German Cancer Research Center (DKFZ), Heidelberg and German Cancer Consortium (DKTK) Partner Site Dresden, Dresden, Germany

Correspondence: frank.buchholz@tu-dresden.de

different regions around the transcription start site (Yeo et al, 2018). Depending on the cellular context, MeCP2 can function as both an activator and a repressor by diverse interactions with transcriptional activators or inhibitors (Chahrour et al, 2008). For its repressive function, MeCP2 binds to methylated CpG dinucleotides and recruits a set of histone modifiers including DNA methyltransferases (DNMT1 and DNMT3A), histone methyltransferases and the SIN3A-histone deacetylase corepressor complex to initiate gene repression (Nan et al, 1998; Fuks et al, 2003; Kimura & Shiota, 2003; Rajavelu et al, 2018). The dCas9-KRAB-MeCP2 fusion protein outperformed both KRAB and MeCP2 as single fusions to dCas9, suggesting that the combination of these two domains leads to a synergistic reinforcement of gene repression (Yeo et al, 2018).

Gene inactivation mediated by CRISPRi is thought to be transient in nature, with transcription rapidly recovering once the fusion protein is released from its binding site (Amabile et al, 2016; Nuñez et al, 2021). To establish long-term memory of gene silencing, a panel of epigenome editing tools including the CRISPRoff and the Hit-and-run ETR systems have been designed, inspired by the epigenetic silencing of endogenous retroviruses (ERVs) (Amabile et al, 2016; Stepper et al, 2017; Nuñez et al, 2021). Here, dCas9 was fused to the KRAB domain of KOX1 in combination with the catalytic domains of the DNA methyltransferases DNMT3A and DNMT3L (Amabile et al, 2016; Stepper et al, 2017; Nuñez et al, 2021). Transient expression of these repressors initiated highly specific DNA methylation and gene repression that was maintained through cell division and differentiation (Amabile et al, 2016; Nuñez et al, 2021). Moreover, the targeted repression could be readily switched "on" and "off" from the same promoter via iterative cycles of DNA methylation and demethylation (Amabile et al, 2016; Nuñez et al, 2021).

In this study, we unexpectedly find that dCas-KRAB-MeCP2 can also induce long-term epigenetic gene repression. Transient expression of this cassette led to long-term gene silencing in mouse embryonic stem cells (mESCs), accompanied by elevated DNA methylation at the target gene promoter. Gene silencing was maintained for more than 1 mo in cell culture and persisted following embryoid body (EB) differentiation. Surprisingly, however, DNMT3A and DNMT3B were not required to achieve long-term silencing. In fact, long-term gene silencing was achieved and even enhanced in DNMT3A/3B null cells, suggesting the establishment of a long-term silencing state independent of DNA methylation. To improve the long-term silencing of this epigenetic silencing tool, we optimized the fusion protein and developed transient delivery methods for use in human and mouse cells.

## Results

### dCas9-KRAB-MeCP2 induces long-term gene repression in mESCs

The dCas9-KRAB-MeCP2 cassette has been shown to induce transient gene repression in a panel of cell lines (Yeo et al, 2018). To investigate the potential application and mechanism of this gene silencer in mESCs, we constructed a doxycycline inducible dCas9-KRAB-MeCP2 vector and compared it with a doxycycline inducible

dCas9-KRAB version. Both plasmids were, respectively, transfected into Oct4-GFP mESCs (Ying et al, 2003), in which GFP expression is driven by the Oct4 promoter. Hygromycin B was applied to select for resistant cells that had stably integrated the expression cassettes (Fig 1A). dCas-KRAB-MeCP2 can efficiently silence gene expression when it is targeted to the transcription start site within a 1 kb up- or down-stream window (Yeo et al, 2018). Therefore, we designed three chemically synthesized sgRNAs targeting the N-terminal coding sequence of GFP within this range (sgGFPs, Table S1), and transiently transfected the sgRNA pool to evaluate gene silencing efficiencies. After sgRNA transfection, doxycycline was applied to induce dCas9-KRAB-MeCP2 or dCas9-KRAB expression for 2 d. To quantify gene silencing, GFP expression was measured by flow cytometry (FACS) before (day 0) and after sgRNA transfection on day 3 and day 10. In cells expressing dCas9-KRAB, mild GFP repression (17.4%) was measured 3 d post sgGFP transfection, which decreased rapidly to non-detectable level at day 10, suggesting that dCas9-KRAB induced transient gene repression (Fig 1B and C). In the dCas9-KRAB-MeCP2 sample, higher levels of GFP repression was observed (78.1%) 3 d post sgGFP transfection, confirming that dCas9-KRAB-MeCP2 is more potent than dCas9-KRAB in repressing gene expression (Fig 1B and D) (Yeo et al, 2018). Interestingly, substantial GFP repression (69.8%) was maintained 10 d post sgGFP transfection, indicating that the repression was longer lasting in comparison with dCas9-KRAB. To investigate how long this effect could last, we kept growing the cells in culture. Surprisingly, the GFP depletion lasted for many cell passages in the dCas9-KRAB-MeCP2 sample, with 64.1% GFP repression quantified 30 d post sgRNA transfection, suggesting that expression of dCas9-KRAB-MeCP2 induced long-term GFP silencing in mESCs (Fig 1B and D). dCas9-KRAB-MeCP2 constructs have previously been used for transient CRISPRi experiments in other cell types, but long-lasting gene suppression has not been reported (Yeo et al, 2018). To test if the long-term silencing effect is promoter specific, we generated a Gapdh-GFP reporter vector and stably integrated this construct into mESCs carrying the inducible dCas9-KRAB-MeCP2 cassette (Fig 2A). 67% of this cell pool expressed GFP. This percentage did not change after transfection with a non-targeting sgRNA and culturing of the cells for 30 d (Fig 2B, sgControl, Table S1). In sharp contrast, only 1% GFP positive cells were measured 30 d post transfection with the sgRNA targeting GFP (Fig 2B), indicating that long term silencing by dCas9-KRAB-MeCP2 in mESCs is not promoter specific.

mESCs can be differentiated into EBs to initiate lineage-specific differentiation into the three germ layers endoderm, ectoderm and mesoderm (Wang & Yang, 2008). To investigate, whether cell differentiation influences dCas9-KRAB-MeCP2–mediated long-term gene silencing, we differentiated the sgGFP silenced Gapdh-GFP cells into EBs. 10 d after in vitro hanging drop EB differentiation, cells with various differentiated morphologies and spontaneous cardiomyocyte contractions were observed, demonstrating efficient differentiation of the mESCs (Fig 2C) (Wang & Yang, 2008). Remarkably, GFP silencing persisted after differentiation of EBs, with only 0.1% GFP-positive cells quantified in the EB sample differentiated from mESCs treated with the sgGFP (Fig 2C). This result suggests that long-term gene silencing induced by dCas9-KRAB-MeCP2 is maintained during EB differentiation.

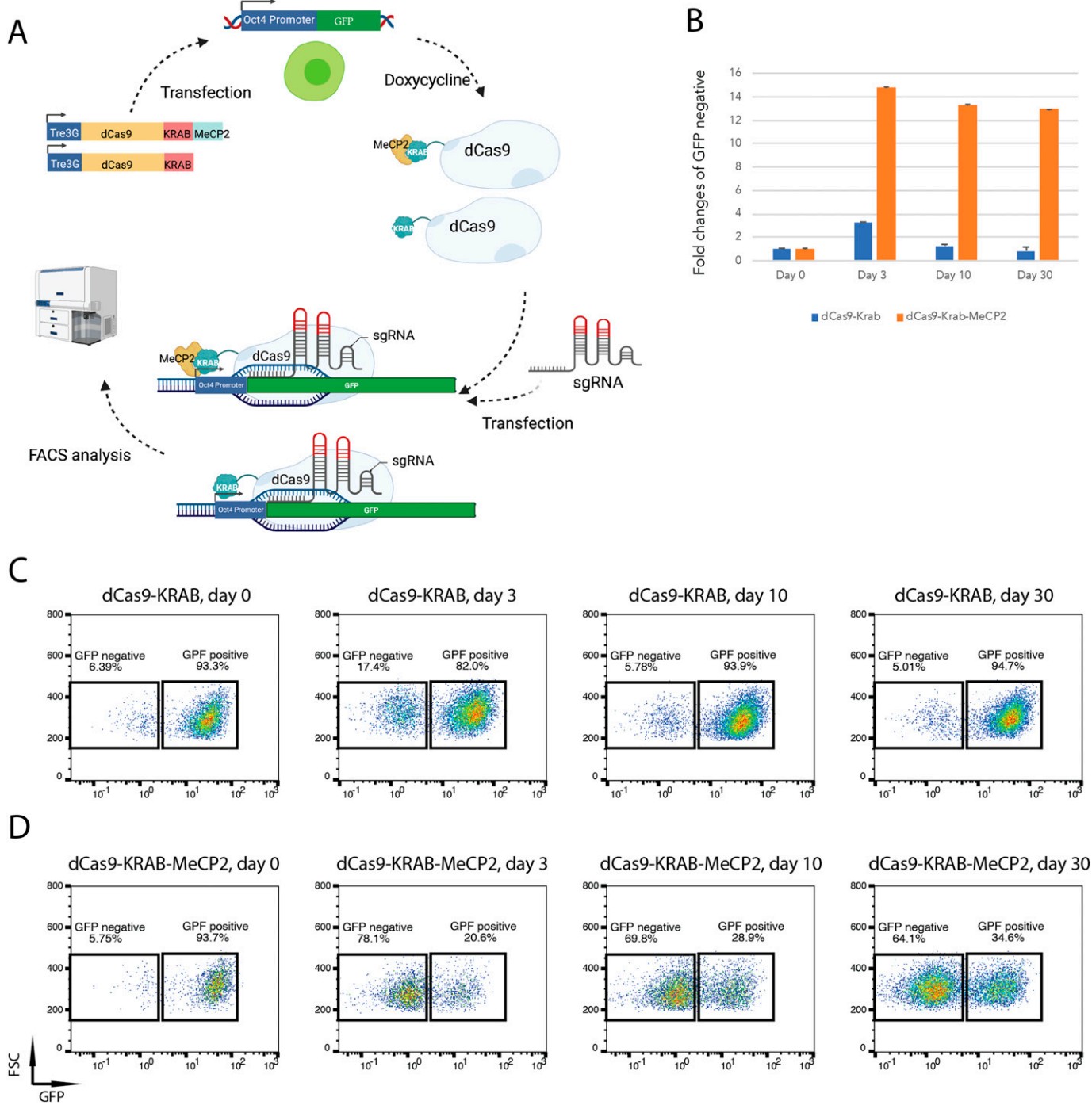

**Figure 1. dCas9-KRAB-MeCP2 mediates persistent gene silencing.**
**(A)** Flow diagram illustrating gene repression by dCas9-KRAB or dCas9-KRAB-MeCP2 in Oct4-GFP mouse embryonic stem cells. Important steps are shown by arrows. dCas9-KRAB or dCas9-KRAB-MeCP2 stably transfected into Oct4-GFP mouse embryonic stem cells were induced by doxycycline and targeted to the Oct4 promoter by short guide RNAs (sgRNAs) to silence GFP expression. The Tre3G promoter of Tet-On 3G transactivator system drives the expression of dCas9-fusion proteins in the presence of doxycycline. **(B)** Quantification of GFP repression by dCas9-KRAB or dCas9-KRAB-MeCP2 over time. Fold changes of GFP repression after sgRNAs transfection in dCas9-KRAB or dCas9-KRAB-MeCP2 samples at denoted time points, normalized to samples before sgRNA transfection (Day 0) are shown. Data are represented as mean ± SD from three independent replicates. **(C, D)** Flow cytometry plots of dCas9-KRAB (C) or dCas9-KRAB-MeCP2 (D) mediated GFP silencing at denoted time points in a 30-d time course. The percentages of GFP positive versus GFP negative cells are indicated in the boxes. FSC, forward scatter.

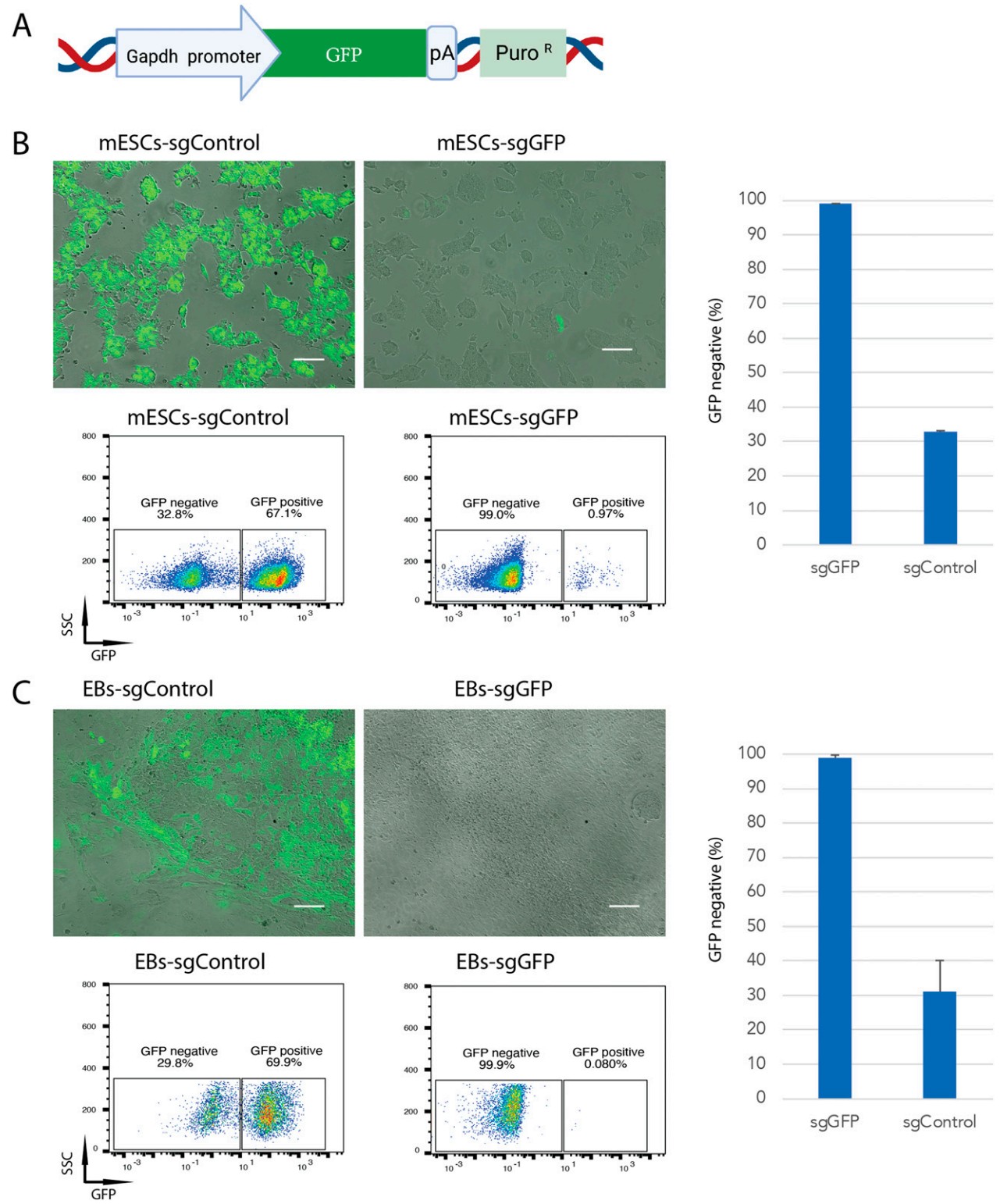

**Figure 2. Gene repression mediated by dCas9-KRAB-MeCP2 is not promoter specific and is maintained during embryoid body differentiation.**
**(A)** Schematic presentation of the transgene stably integrated into mouse embryonic stem cells (mESCs). Important elements are indicated. pA, simplex virus thymidine kinase polyadenylation signal, PuroR, puromycin resistance gene. **(B)** Representative microscope images (top panels) and flow cytometry plots (bottom panels) of GFP expression in Gaphd-GFP mESCs transfected with sgGFP (right) or sgControl (left) 30 d post transfection. The percentages of GFP positive versus GFP negative cells are indicated in the boxes. FSC, forward scatter. Scale bars are 100 μm. Data are represented as mean ± SD from three independent replicates (right graph). **(C)** Representative microscope images (top panels) and flow cytometry plots (bottom panels) of GFP expression in embryoid bodies differentiated from Gaphd-GFP mESCs

## dCas9-KRAB-MeCP2 induces long-term gene repression in human HEK293 cells

To determine if the long-term silencing effect can be established in other cell types, we generated doxycycline inducible dCas9-KRAB and dCas9-KRAB-MeCP2 HEK293 cell lines, where a Gapdh-GFP cassette was stably integrated to report GFP expression (Fig 2A). As expected, transient transfection of sgGFP induced GFP repression in dCas9-KRAB-MeCP2 and dCas9-KRAB Gapdh-GFP HEK293 cells. To investigate the duration of GFP silencing, we collected GFP negative cells by FACS 5 d post transfection of sgGFP and monitored GFP expression of these two samples over a period of 30 d (Fig S1A). As in the mES cells, HEK293 cells transfected with dCas9-KRAB lost GFP repression gradually over time. Approximate 75% of the cells regained GFP expression 10 d after cells sorting, which increased to more than 90% at the end of experiment (30 d after sorting, Fig S1B). In contrast, GFP negative percentages of the dCas9-KRAB-MeCP2 sample remained unchanged for the whole period of cell culture, indicating that dCas9-KRAB-MeCP2 can induce long-term silencing in HEK293 cells (Fig S1B).

## Optimization of the silencing cassette

To investigate whether long-term gene silencing can be improved, we generated several constructs and compared them with the original version. In addition to MeCP2, Yeo and colleagues reported another methyl-CpG–binding domain (MBD) protein, MBD2B, as a strong transcription repressor when fused to dCas9 (Yeo et al, 2018). To determine if we can improve gene silencing efficiency and induce long-term gene repression using this protein, we assembled a doxycycline inducible dCas9-KRAB-MBD2B construct and evaluated its potency for gene repression in Oct4-GFP mESCs in comparison with the original dCas9-KRAB-MeCP2 (Fig S2A). FACS analysis showed a much weaker GFP repression by dCas9-KRAB-MBD2B (20.8%) compared with dCas9-KRAB-MeCP2 (71.1%) 3 d post sgGFP transfection (Fig S2B). More importantly, dCas9-KRAB-MBD2B induced only a transient repression, with GFP expression restored to wild-type levels 10 d after sgGFP transfection (Fig S2B). This result suggests that the MeCP2 domain is required for log-term silencing and that it cannot be replaced by the methyl-CpG-binding domain of MBD2B.

A recent screen of KRAB domain proteins identified the ZIM3 KRAB domain as a potent transcriptional repressor, with dCas9-ZIM3 outperforming dCas9-KRAB fusions in gene silencing (Alerasool et al, 2020). To determine if the ZIM3 outperforms the Kox1 KRAB domain for long-term gene repression, we generated doxycycline inducible dCas9-ZIM3 and dCas9-ZIM3-MeCP2 constructs and compared them with dCas9-KRAB-MeCP2 for GFP repression in Oct4-GFP mESCs (Fig S2A). Indeed, much stronger GFP repression was observed 3 d post transfection for the dCas9-ZIM3 construct compared with the dCas9-KRAB-MeCP2 construct (87.6% versus 77.1%, Fig S2C). Furthermore, GFP silencing was also increased for the dCas9-ZIM3-MeCP2 version from 87.6 to 92.5%

(Fig S2C), suggesting that replacing the Kox1 KRAB domain with the ZIM3 KRAB domain improves long-term gene repression.

mRNA delivery into cells is a convenient form for transient expression of proteins and recent advances have overcome some innate obstacles for mRNA applications such as short half-life and unfavorable immunogenicity. These advances have extended the therapeutic potential of mRNAs for a wide range of applications including vaccines, cancer immunotherapies, cellular reprogramming, and genome editing (Hou et al, 2021). To determine potential mRNA applications of the different silencing cassettes for gene repression, we synthesized in vitro transcribed (IVT) dCas9-KRAB-MeCP2 and dCas9-ZIM3-MeCP2 mRNAs and co-transfected them, respectively, with sgGFP into Oct4-GFP mES cells (Fig 3A). 2.3% GFP negative cells in the sgControl sample and 22% in the sgGFP sample were measured after transfection with dCas9-KRAB-MeCP2 mRNA, suggesting that mRNA delivery can induce gene repression (Fig 3B). Prominent improvements of gene repression were observed using dCas9-ZIM3 (34%) or dCas9-ZIM3-MeCP2 (63%) mRNA, respectively, 3 d post mRNA transfection (Fig 3C). Therefore, ZIM3 synergizes with MeCP2 in that dCas-ZIM3-MeCP2 outperformed dCas9-ZIM3 and dCas9-KRAB-MeCP2 to repress gene expression. Incorporation of nucleotide analogs during mRNA synthesis can improve the properties of IVT mRNA both in terms of stability and decreased immunogenicity (Warren et al, 2010). We synthesized dCas9-KRAB-MeCP2 mRNA incorporated with nucleotide analogs of 5mCTP and Pseudo-UTP, and transfected the modified mRNA into Oct4-GFP mESCs to determine the activity for gene repression (Fig 3A). Incorporation of the modified nucleotide significantly improved gene silencing with GFP repression increased to 67% in Oct4-GFP mESCs (Fig 3D). To maximize the silencing capacity for mRNA transfections, we synthesized dCas9-ZIM3-MeCP2 mRNA with 5mCTP and Pseudo-UTP incorporation. With these measures, we achieved 89% GFP repression, which was maintained for more than 30 d, suggesting that delivery of dCas9-ZIM3-MeCP mRNA with modified nucleotides can induced highly efficient and persistent gene repression (Fig 3D and E).

To determine whether dCas9-ZIM3-MeCP2 mRNA can efficiently silence gene expression in primary human cells, we tested the approach in primary human T cells. We designed sgRNAs targeting the *CD8* promoter (sgCD8, Table S1) to silence CD8 expression and co-electroporated this sgRNA together with 5mCTP and Pseudo-UTP modified dCas9-ZIM3-MeCP2 mRNA into CD8[+] primary T lymphocytes. To quantify gene silencing efficiency, the CD8 protein was stained with an APC conjugated antibody, and quantified by FACS. Transfection of dCas9-ZIM3-MeCP2 mRNA and sgCD8 induced CD8 silencing in primary T lymphocytes with 11.5% efficiency 7 d after electroporation (Fig 3F). We conclude that dCas9-ZIM3-MeCP2 delivered as an mRNA can silence gene expression in primary cells.

## dCas9-KRAB-MeCP2 induces DNA methylation at sgRNA target sites

Epigenetic modifications including histone modification (e.g., H3K9 and H3K27 methylation) and DNA methylation can induce long-term

---

transfected with sgGFP (right) or sgControl (left). The percentages of GFP positive versus GFP negative cells are indicated in the boxes. FSC, forward scatter. Scale bars are 100 µm. Data are represented as mean ± SD from three independent replicates (right graph).

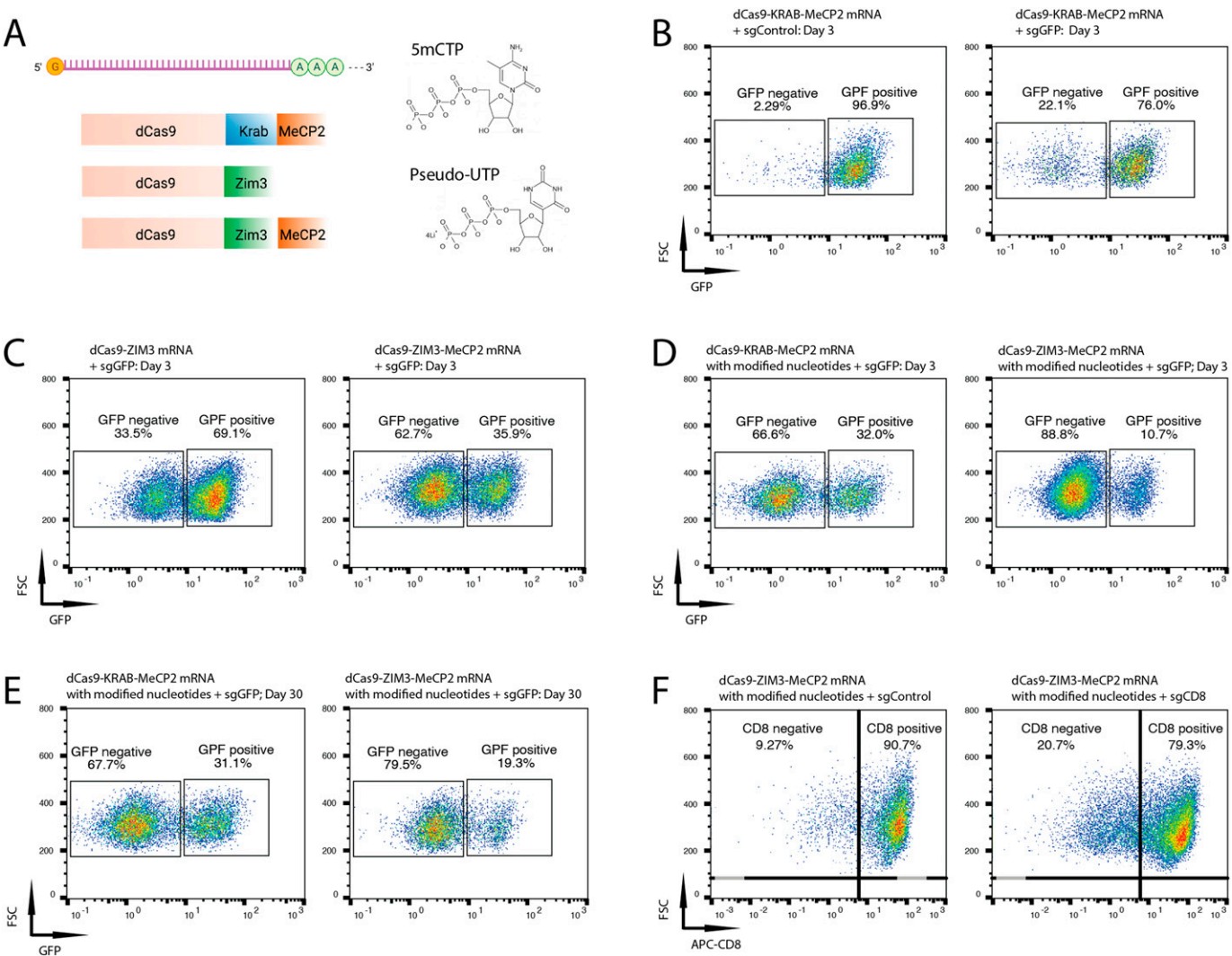

**Figure 3. Optimization of the silencing cassettes for gene repression.**
**(A)** Schematics of in vitro transcription of dCas9-KRAB-MeCP2, dCas9-ZIM3 and dCas9-ZIM3-MeCP2 mRNAs with modified nucleotides (5mCTP and pseudo-UTP).
**(B)** Representative flow cytometry plots of Oct4-GFP mouse embryonic stem cells (mESCs) transfected with dCas9-KRAB-MeCP2 mRNA and sgControl (left) or sgGFP (right).
**(C)** Representative flow cytometry plots of Oct4-GFP mESCs transfected with sgGFP and dCas9-ZIM3 mRNA (left) or dCas9-ZIM3-MeCP2 mRNA (right). **(D, E)** Representative of flow cytometry plots of Oct4-GFP mESCs transfected with sgGFP and dCas9-KRAB-MeCP2 mRNA (left) or dCas9-ZIM3-MeCP2 mRNA (right) with modified nucleotides at indicated time points. **(F)** Representative of flow cytometry plots of CD8+ T lymphocytes transfected with dCas9-ZIM3-MeCP2 mRNA with modified nucleotides and sgControl (left) or sgCD8 (right).

silencing of proximal genes (Yeo et al, 2018; Nuñez et al, 2021). Inspired by a report that MeCP2 physically interacts and regulates the activity of DNA methyltransferase 3A (DNMT3A) (Rajavelu et al, 2018), we determined the DNA methylation status of the GFP promoter in the Oct4-GFP mESCs through bisulfite sequencing and investigated its causality to the long-term gene silencing effects (Nuñez et al, 2021). Bisulfite sequencing revealed a pronounced increase of DNA methylation at CpG islands of the Oct4-GFP sequence in the doxycycline inducible dCas9-KRAB-MeCP2 samples transiently transfected with sgGFP (Fig 4A). To determine if dCas9-KRAB-MeCP2 can also induce DNA methylation of endogenous genes, we designed sgRNA targeting the mouse *Fgf5* promoter (sgFgf5, Table S1). Again, transfection of sgFgf5 markedly induced DNA methylation at CpG islands of the *Fgf5* promoter in doxycycline

induced dCas9-KRAB-MeCP2 mESCs (Fig 4B), demonstrating that dCas9-KRAB-MeCP2 can induce DNA methylation of endogenous promoters in mESCs.

DNA methylation at promoters has been associated with persistent transcriptional repression (Amabile et al, 2016; Liu et al, 2016; Nuñez et al, 2021). To determine if dCas9-KRAB-MeCP2 can induce persistent repression of a gene lacking a canonical CpG island, we designed an sgRNA targeting to mouse *lefty1* promoter (sgLefty1, Table S1), which does not contain CpG islands in its promoter region (Gardiner-Garden & Frommer, 1987). Consistent with a previous report that *lefty1* is a nonessential gene for mESCs, we did not observe cell variability phenotype after transient transfection of sgLeft1 and dCas9-KRAB-MeCP2 mRNA with modified nucleotides (Meno et al, 1998). Nevertheless, Lefty1 expression was strongly and

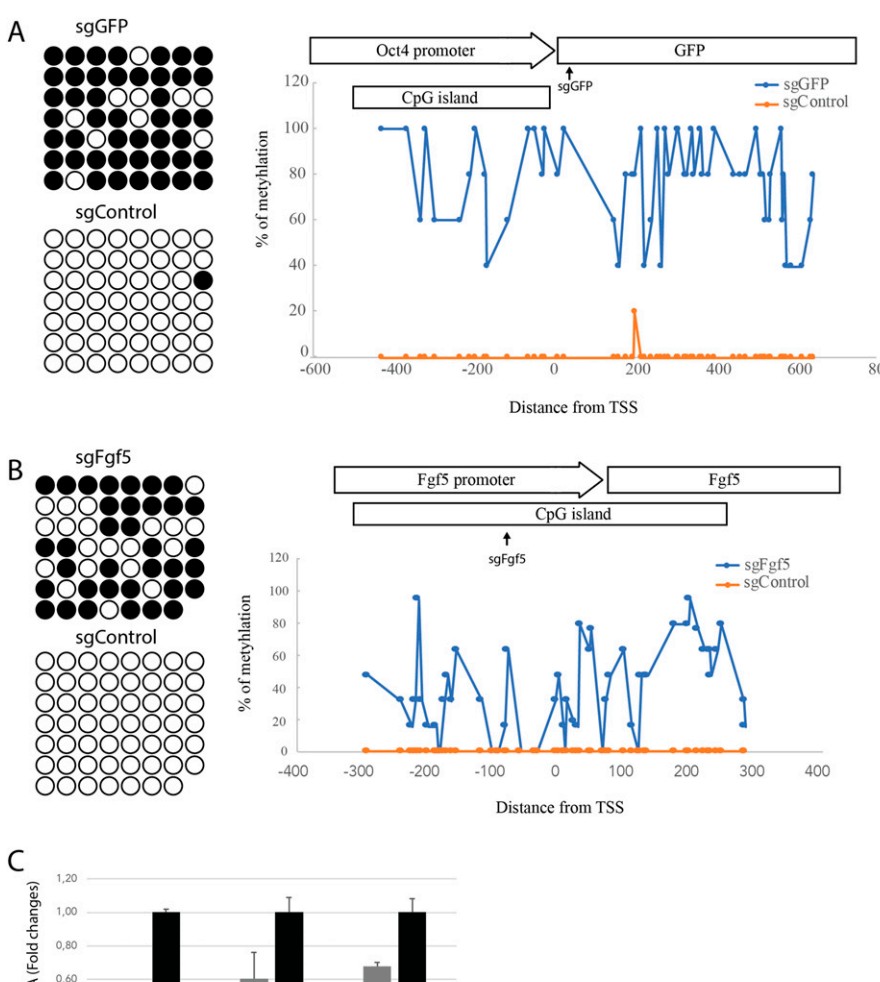

**Figure 4. dCas9-KRAB-MeCP2 induced DNA methylation at short guide RNA target sites.**
**(A)** Bisulfite sequencing analysis of DNA methylation at Oct4-GFP locus after transfection of sgGFP or sgControl (left panel). The white circles indicate unmethylated CpG dinucleotides and black circles represent methylated CpG dinucleotides. Bisulfite quantification of DNA methylation of sgGFP and sgControl samples, at the Oct4-GFP locus depicted in the top schematic (right panel). Data show percentage of CpG methylation. **(B)** Bisulfite sequencing analysis of DNA methylation at the Fgf5 locus after transfection of sgFgf5 or sgControl (left panel). Bisulfite quantification of DNA methylation of sgFgf5 and sgControl samples, at Fgf5 locus depicted in the top schematic (right panel). **(C)** Time course quantification of Lefty1 RNA expression after transfection of sgLefty1 and dCas9-KRAB-MeCP2 mRNA by qRT-PCR. Lefty1 expression was normalized to Gapdh. Data are represented as mean ± SD from three independent replicates.

persistently repressed, with expression levels reaching 49%, 60% and 68% of normal after 7, 14 and 21 d, respectively, when the cells were transiently transfected with sgLefty1 and dCas9-KRAB-MeCP2 mRNA (Fig 4C). Hence, transient transfection of dCas9-KRAB-MeCP2 can induce persistent repression of a non-CpG island promoter gene.

### DNA methylation is dispensable for dCas9-KRABd-MeCP2–mediated gene repression

Based on the obtained results, we assumed that the persistent gene silencing was mediated by DNA methylation at the sites targeted. We therefore decided to investigate the potential involvement of DNA methyltransferases (DNMTs) in the silencing process. There are three DNMTs that regulate DNA methylation in mammals. DNMT1 is responsible for maintaining methylation patterns following DNA replication and shows a preference for hemi-methylated DNA. In contrast, DNMT3A and DNMT3B are essential for

de novo methylation, in particular during early embryogenesis (Aguilera et al, 2010; Liao et al, 2015). Hence, DNMT3A and/or DNMT3B are likely candidates contributing to the observed DNA methylation. To determine which of the two de novo methylation DNMT3s participate in dCas9-KRAB-MeCP2–mediated DNA methylation, we performed CRISPR/Cas9–mediated knockouts by co-transfection of a CRISPR/Cas9 plasmid together with dual sgRNAs targeting *DNMT3A* or *DNMT3B* into Oct4-GFP mESCs (Table S1) (Chen et al, 2014). Homozygous *DNMT3A* and *DNMT3B* knockout clones were isolated, and confirmed by PCR amplification of the truncated fragments (Fig S3A). *DNMT3A* or *DNMT3B* knockout clones were further validated by Western blot hybridization using specific antibodies (Fig 5A). If one of the DNMTs was involved in the long-term silencing, we expected to see that now the transfection of sgGFP would only results in transient silencing of GFP expression. Surprisingly, transient transfection of dCas9-KRAB-MeCP2 and sgGFP into either *DNMT3A* or *DNMT3B* knockout clones still resulted in long-term silencing of GFP (Figs 5B and S3B). DNMT3A and DNMT3B proteins physically associate

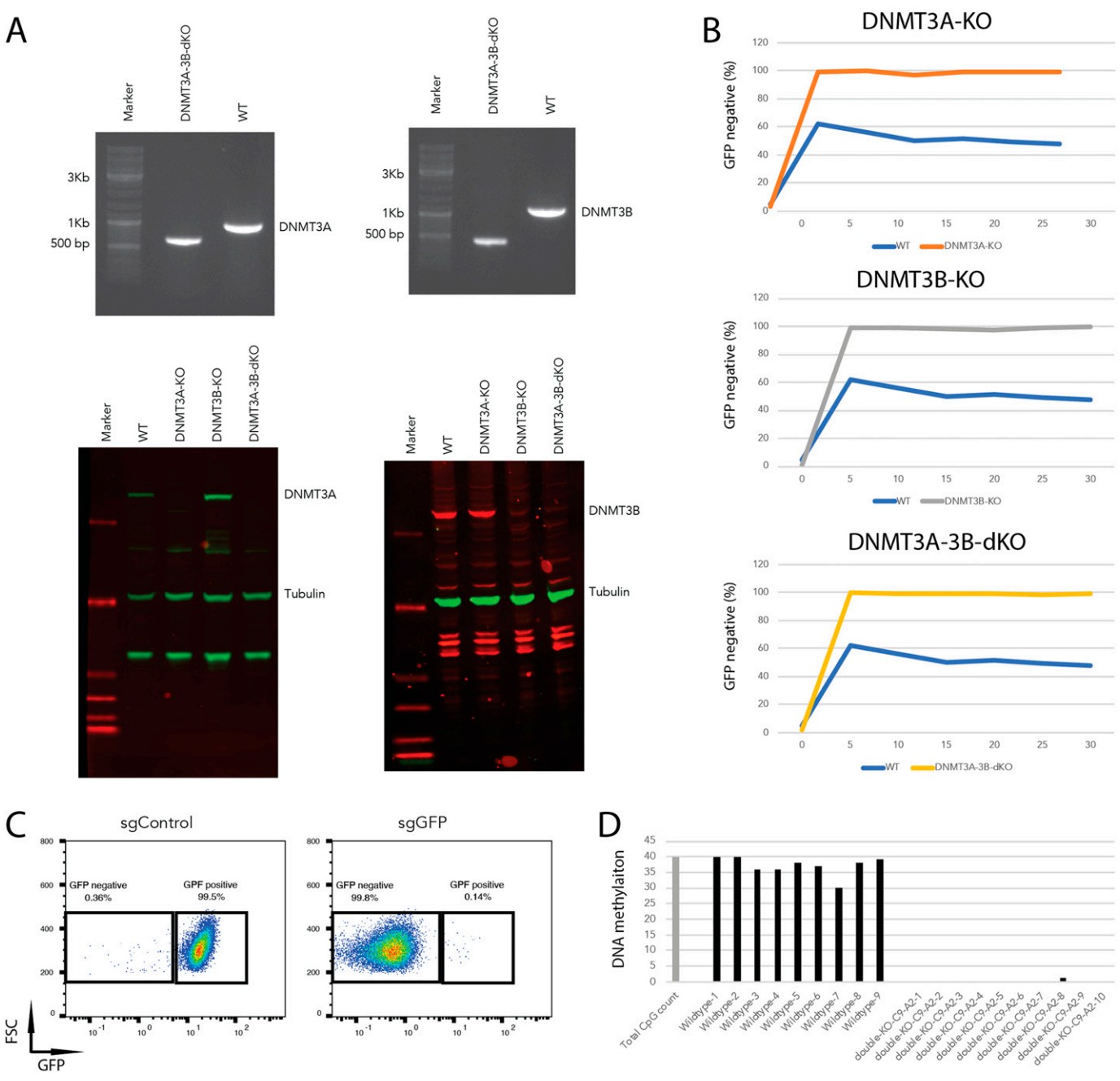

**Figure 5. DNA methylation is dispensable for dCas9-KRABd-MeCP2–mediated gene repression.**
**(A)** PCR amplification (top panels) and Western blot hybridization (bottom panels) to determine DNMT3A (left) and DNMT3B (right) single or double knockouts in mouse embryonic stem cells (mESCs). Anti-DNMT3A and anti-DNMT3B antibodies were used to visualize DNMT3A (102 kD, green channel) and DNMT3B (96 kD, red channel) proteins. An anti-tubulin antibody was used to detect tubulin (50 kD, green channel) as the loading control. **(B)** Persistent GFP repression mediated by dCas9-KRAB-MeCP2 in DNMT3A knockout (top), DNMT3B knockout (middle), and DNMT3A-3B double knockout (bottom) Oct4-GFP mESCs. GFP repression after transient short guide RNA transfection (Day 0) of individual samples were determined at denoted time points. **(C)** Representative flow cytometry plots of DNMT3A-3B double knockout Oct4-GFP mESCs transfected with sgControl (left) or sgGFP (right) 30 d after short guide RNA transfection. The percentages of GFP positive versus GFP negative cells are indicated in the boxes. FSC, forward scatter. **(D)** Bisulfite sequencing analysis of DNA methylation at Oct4-GFP locus in the wild type and DNMT3A-DNMT3B double knockout Oct4-GFP mESCs after transfection of sgGFP. The grey bar illustrates the possible number of CpG dinucleotides in the investigated genomic region.

in mESCs, and appear to act synergistically to methylate promoters during mESC differentiation (Li et al, 2007). It was therefore possible that single knockouts of *DNMT3A* or *DNMT3B* may not completely abolish DNA methylation, thereby maintaining long-term

gene silencing. To investigate this possibility, we generated *DNMT3A/DNMT3B* double knockout mESCs clones by transfection of a Cas9 plasmid together with dual sgDNMT3A into *DNMT3B* knockout cells. The successful double knockout of *DNMT3A* and *DNMT3B* was

confirmed by amplification of truncated PCR products and Western blot hybridization (Fig 5A). To our surprise, also the double knockout cells showed the persistent GFP silencing phenotype when the cells were transfected with sgGFP (Fig 5B). Furthermore, transfection of dCas9-KRAB-MeCP2 and sgGFP into *DNMT3A/DNMT3B* single or double knockout cells induced even more efficient long-term silencing of GFP than in the wild type mESCs, with >99% of the cells maintaining GFP negative over a culture period of 30 d (Figs 5C and S3B). Importantly, bisulfite sequencing revealed that DNA methylation at the sgGFP target region was completely abolished in the double knockout samples, demonstrating that DNA methylation is dispensable for long-term gene silencing in these cells (Fig 5D). Hence, DNA methylation is not required to achieve long-term silencing using this method. To start investigating other possible mechanisms of long-term dCas9-KRAB-MeCP2–mediated gene silencing, we turned to putative repressive histone modifications and investigated H3K9 methylation because this repressive histone modification has been previously associated with MeCP2 binding (Thambirajah et al, 2012). Indeed, chromatin immunoprecipitation (ChIP) using an anti-H3K9me3 antibody revealed significant increase in H3K9me3 at the dCas9-KRAB-MeCP2 target region in mESCs, suggesting that long-term gene silencing might be maintained by repressive histone modifications (i.e., H3K9me3, Fig S4).

# Discussion

Here we have reported that transient sgRNA mediated targeting of dCas9-KRABd-MeCP2 can induce persistent gene silencing and DNA methylation at target promoters. To dissect the causality between DNA methylation and the long-term gene repression, we generated *DNMT3A/DNMT3B* knockout cell lines. Although DNA methylation at sgRNA-targeted regions was completely abolished, transient transfection of dCas9-KRABd-MeCP2 still induced long-term gene silencing with even enhanced efficiency in these cells, suggesting that persistent gene silencing mediated by dCas9-KRABd-MeCP2 is independent of DNA methylation. Our analysis therefore suggests that the mechanism behind this effect is divergent from the recently described KRAB-DNMT3A-DNMT3L systems (Amabile et al, 2016; Stepper et al, 2017). For the KRAB-DNMT3A-DNMT3L systems, establishment of persistent gene repression is initiated by DNA methylation and maintained by the orchestration of repressive histone modification (H3K9me3) and DNA methylation. Therefore, DNA methylation is essential for establishment and maintenance of persistent gene repression for these epigenetic silencers (Nuñez et al, 2021). This is not the case for dCas9-KRABd-MeCP2. MeCP2 is an epigenetic repressor bridging DNA methylation and histone modifications to convert epigenetic silencing signals to transcriptional repression. The MBD of MeCP2 recognizes and binds to methylated DNA, whereas the TRD interacts with chromatin remodeling proteins to induce inaccessible chromatin conformations for the transcription machinery (Ragione et al, 2016; Rajavelu et al, 2018). Thus, DNA methylation appears not to be a prerequisite for MeCP2-mediated gene repression, in line with our observation that dCas9-KRABd-MeCP2

can induce persistent gene repression in the absence of DNA methylation. The exact mechanism of persistent repression mediated by dCas9-KRABd-MeCP2 has to be investigated in the future, but first results hint at histone tail modifications as a possible process implicated in this context. Nevertheless, the developed methods for transient delivery of dCas9-KRABd-MeCP2 activity to persistently silence genes in mammalian cells presented here should provide a useful instrument in the ever-growing CRISPR tool box.

# Materials and Methods

## Plasmids and sgRNAs cloning

Doxycycline-inducible dCas9-KRAB and dCas9-KRAB-MeCP2 plasmids were constructed by replacing the dCas9-VRP fragment of PB-TRE-dCas9-VPR plasmid (63800; Addgene) with dCas9-KRAB or dCas9-KRAB-MeCP2 fragments from Addgene plasmid (110821). The ZIM3 fragment was amplified from Addgene plasmid (154472). dCas9-ZIM3 and dCas9-ZIM3-MeCP2 plasmids were constructed by replacing the KRAB fragment of doxycycline inducible dCas9-KRAB and dCas9-KRAB-MeCP2 plasmids with ZIM3. CRISPR/Cas9 plasmid px459V2.0 (134451; Addgene) was used for cloning sgRNAs and generation of DNMT knockouts. Chemically synthesized sgRNAs were ordered from Synthego Corporation. Sequences of gRNAs are listed in Table S1.

## IVT mRNA synthesis

dCas9-KRAB-MeCP2, dCas9-ZIM3, and dCas9-ZIM3-MeCP2 cassettes were amplified from the doxycycline inducible plasmids using Herculase II Fusion DNA Polymerase (Agilent) and primers listed in Table S1. *IVT* mRNAs were transcribed from the purified PCR products using HiScribe T7 ARCA mRNA Kit (NEB) with/without 5mCTP and Pseudo-UTP (TriLink Biotechnologies) incorporation, and purified using Monarch RNA Cleanup Kit (NEB).

## Cell culture and transfection

mESCs and HEK293 cells were cultured in Glasgow Minimum Essential Medium (GMEM; Sigma-Aldrich), supplemented with 10% FBS (Pan-biotech), 1,000 U/ml LIF (ESGRO), 100 mM nonessential amino acids (Invitrogen), 2 mM L-glutamine. Lipofectamine 2000 (Thermo Fisher Scientific) was used to transfect plasmids into mESCs and HKE293 cells. Lipofectamine MessengerMax (Thermo Fisher Scientific) was used to transfect mRNA and chemically synthesized sgRNAs into mESCs and HEK293 cells. GFP expression was analyzed by fluorescence microscope imaging (EVOS; Thermo Fisher Scientific) or by FACS (MACSQuant X; Miltenyi Biotec).

Primary T lymphocytes were prepared from healthy donors as previously described with minor modifications (Sreevalsan et al, 2020). Whole blood samples were collected in EDTA and PBMCs were isolated by density gradient centrifugation using Ficoll-Paque PLUS (GE Healthcare). Subsequently, CD8[+] T cells were subjected to negative selection using the CD8[+] T Cell Isolation kit (MACS; Miltenyi

Biotec), expanded and activated using Dynabeads Human T-Activator CD3/CD28 (Gibco) in ImmunoCult-XF T Cell Expansion Medium (Stemcell Technologies). Activation beads were removed after 72 h and CD8⁺ T cells were kept in culture for 24 h before functional tests were performed. 1 µg dCas9-KRAB-MeCP2 mRNA with incorporation of modified nucleotides was nucleofected into one million activated T cells along with 0.2 µg sgCD8 or sgControl (P3 Primary Cell 4D-NucleofectorTM X Kit S, program CM138; Lonza). 1 wk after nucleofection, cells were stained with CD8 APC-Cy7 antibody (348813; BD Pharmingen) and analyzed by flow cytometry (MACSQuant X; Miltenyi Biotec).

### Bisulfite sequencing PCR

DNA methylation of sgRNA target region was determined by bisulfite sequencing PCR as previously described with modifications (Nuñez et al, 2021). Genomic DNA isolation, bisulfite conversion, and cleanup were performed from $1 \times 10^4$ cells according to manufacturer's instructions using the EZ DNA Methylation-Direct kit (ZYMO Research). Purified bisulfite-converted DNA was amplified using EpiMark Hot Start Taq (NEB). Amplicons were cloned into the pCR2.1 vector according to manufacturer's instructions using the TA Cloning Kit (Invitrogen). Colonies were picked and sequenced by Sanger sequencing. Primer sequences used for bisulfite-PCR amplification are listed in Table S1.

### CRISPR/Cas9–mediated knockout of DNMTs

DNMTs knockouts were generated by transfection of dual sgRNAs on the Addgene plasmid px459 that target to DNMTs into mESCs. sgRNAs target sequences are listed in Table S1. Individual homozygous clones for DNMT3A and DNMT3B knockout mESCs were isolated. DNMTs knockouts were confirmed by Sanger sequencing of the truncated PCR fragments using primers listed in Table S1, and Western blot hybridization.

### Western blot hybridization

$5 \times 10^5$ mESCs were harvested and lysed in Laemmli sample buffer. 10 µg of protein extracts were separated on NuPAGE 4–12% Bis-Tris protein gels (Invitrogen) and blotted to nitrocellulose membrane (Millipore). The membranes were probed with primary antibodies against DNMT3A (sc-365769; Santa Cruz), DNMT3B (ab122932; Abcam), and Tubulin (sc-58880; Santa Cruz), and corresponding secondary antibodies (680RD/800CW anti-mouse IgG, and anti-rabbit IgG; RDye). The membranes were scanned by an Odyssey Infrared Imager, and analyzed by the software Image Studio.

### RNA extraction and qRT-PCR analysis

Total RNA was extracted from cells using RNeasy Mini Kit (QIAGEN). 500 ng total RNA was reversed transcribed with Superscript III Reverse Transcriptase (Invitrogen) using an oligo d(T)18 primer. qPCR quantification of gene expression was performed using a real-time thermal cycler (CFX96; Bio-Rad) with primers as shown in Table S1.

### ChIP-qPCR

ChIP experiments were performed as previously described with minor modification (Marks et al, 2012). 4 d after sgGFP transfection, GFP negative cells were collected by FACS. 1 million of sgGFP or sgControl mESCs cells were fixed with 1% (vol/vol) formaldehyde (methanol free) at room temperature for 10 min. Cross-linked chromatin was prepared and sheared using truChIP Chromatin Shearing Kits (Covaris). An antibody against H3K9me3 (ab8898) was used for ChIP experiments. Enrichments of H3K9me3 were determined using a real-time thermal cycler (CFX96; Bio-Rad) with primers as shown in Table S1.

## Supplementary Information

## Acknowledgements

This work was supported by the EU FP7 grant SyBoSS (242129), EU H2020 grant UPGRADE (825825) and a grant by the Deutsche Forschungsgemeinschaft (BU 1400/5-2). We would like to thank Dr. Martin Schneider for providing sgRNA constructs.

### Author Contributions

L Ding: conceptualization, data curation, investigation, methodology, and writing—original draft.
L Schmitt: data curation and investigation.
M Brux: investigation.
D Surun: investigation.
M Augsburg: investigation.
F Lansing: investigation.
J Mircetic: investigation.
M Theis: validation, investigation, and methodology.
F Buchholz: conceptualization, resources, supervision, funding acquisition, project administration, and writing—review and editing.

### Conflict of Interest Statement

The authors declare that they have no conflict of interest.

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
