## [Reviewer comments · Life Science Alliance]

Life Science Alliance

DNA-methylation independent long-term epigenetic silencing with dCRISPR/Cas9 fusion proteins

Frank Buchholz, Li Ding, Lukas Schmitt, Melanie Brux, Duran Sürün, Martina Augsburg, Felix Lansing, Jovan Mircetic, and Mirko Theis

DOI: <https://doi.org/10.26508/lsa.202101321>

Corresponding author(s): Frank Buchholz, Centre for Medical Systems Biology

Review Timeline:	Submission Date:	2021-11-27
	Editorial Decision:	2022-01-31
	Revision Received:	2022-02-16
	Editorial Decision:	2022-02-18
	Revision Received:	2022-02-25
	Accepted:	2022-02-28

Scientific Editor: Novella Guidi

Transaction Report:

January 31, 2022

Re: Life Science Alliance manuscript #LSA-2021-01321-T

Dr. Frank Buchholz
Medical Systems Biology
Medical Systems Biology, UCC, Medical Faculty Carl Gustav Carus
Medical Faculty and University Hospital Carl Gustav Carus, TU Dresden
Dresden 01307
Germany

Dear Dr. Buchholz,

Thank you for submitting your manuscript entitled "DNA-methylation independent long-term epigenetic silencing in mammalian cells with dCRISPR/Cas9 fusion proteins" to Life Science Alliance. The manuscript was assessed by expert reviewers, whose comments are appended to this letter. We, thus, encourage you to submit a revised version of the manuscript back to LSA that responds to all of the reviewers' points.

Thank you for this interesting contribution to Life Science Alliance. We are looking forward to receiving your revised manuscript.

Sincerely,

B. MANUSCRIPT ORGANIZATION AND FORMATTING:

Reviewer #1 (Comments to the Authors (Required)):

1. A short summary of the paper, including description of the advance offered to the field.

Ding et al have developed an epigenetic repressor based on a fusion protein including dead cas9, KRAB module and MeCP2. This work shows effective and persistent repression. Repression has been reported in mESC, Embryoid bodies, 293T cells. The system was further optimized with the better ZIM3 krab domain. This optimized form was highly successful in mESC and demonstrated moderate activity in primary human T cells. Curiously, epigenetic repression of this constructs seems not reduced by reducing promoter methylation which invites very interesting followup studies to further elucidate the repression mechanism.

2. For each main point of the paper, please indicate if the data are strongly supportive. If not, explicitly state the additional experiments essential to support the claims made and the timeframe that these would require.

dCas9-KRAB-MeCP2 seems very well supported with FACS measurement and testing in multiple scenarios including different cell types, embryoid bodies, and promoters.

dCas9-ZIM3-MeCP2 superiority to dCas9-KRAB-MeCP2 seems clear from data provided in Fig 3

Repression connection to methylation is solidly studied by the loss of function studies (DNMT3A and B KO; WB and methylation on promoters). Mutants are well characterized (genotype validated, and expression loss) and tests on the mutants produce a clear result.

3. Lastly, indicate any additional issues you feel should be addressed (text changes, data presentation, statistics etc.).

Regarding figures 1B, 2B, 2C and EV1A all show repression but in different formats ("Fold Changes of GFP negative", "GFP expression (%)", "% of GFP negative", "GFP negative percentage"), perhaps it would be clearer for the reader if the format of the representations gets unified.

Reviewer #2 (Comments to the Authors (Required)):

The current manuscript described a novel Crispr-based method for long term silencing of gene expression in a variety of mammalian cell lines. The authors show that a dCas9-KRAB-MeCP2 constructs targeted to the promoter region of a transgene or endogenous gene can suppress their expression for long times (up to more than one month). They also show that if target gene expression is driven by a CpG island promoter, silencing occurs alongside with CpG methylation, however also non CpG-promoter genes can be silenced by the same construct; moreover, knocking out DNMT3A and 3B does not avert gene silencing. Overall, this suggest that silencing occurs through a mechanisms that does not depend on CpG methylation. I find this tool extremely useful for the community, as a number of studies would strongly profit from such an amenable approach. Overall the study is well performed and controlled, and results are properly interpreted. Given the observed lack of specificity towards CpG island promoters, it would have been nice to test to what extent this cas9 derivative might work as a roadblock for transcription (at least at the beginning of the experiment), for example by utilising different gRNAs targeting more upstream regions of the genes (i.e. not so close to the TSS). Also, the reader is clearly left with the question of how this system works at the mechanistic level. One could think to test histone modifications at the different CpG plus and minus promoters, for example it could be interested to study H3K9 methylation. Overall, while this study falls short in providing the characterisation of the mechanisms behind dCas9-KRAB-MeCP2-mediated silencing, it seems very likely to provide a powerful new tool to the scientific community.

January 31, 2022

Re: Life Science Alliance manuscript #LSA-2021-01321-T

Dr. Frank Buchholz
Medical Systems Biology
Medical Systems Biology, UCC, Medical Faculty Carl Gustav Carus
Medical Faculty and University Hospital Carl Gustav Carus, TU Dresden
Dresden 01307
Germany

Dear Dr. Buchholz,

Thank you for submitting your manuscript entitled "DNA-methylation independent long-term epigenetic silencing in mammalian cells with dCRISPR/Cas9 fusion proteins" to Life Science Alliance. The manuscript was assessed by expert reviewers, whose comments are appended to this letter. We, thus, encourage you to submit a revised version of the manuscript back to LSA that responds to all of the reviewers' points.

Thank you for this interesting contribution to Life Science Alliance. We are looking forward to receiving your revised manuscript.

Sincerely,

- A letter addressing the reviewers' comments point by point.
- An editable version of the final text (.DOC or .DOCX) is needed for copyediting (no PDFs).
- High-resolution figure, supplementary figure and video files uploaded as individual files: See our detailed guidelines for preparing your production-ready images, <https://www.life-science-alliance.org/authors>
- Summary blurb (enter in submission system): A short text summarizing in a single sentence the study (max. 200 characters including spaces). This text is used in conjunction with the titles of papers, hence should be informative and complementary to the title and running title. It should describe the context and significance of the findings for a general readership; it should be written in the present tense and refer to the work in the third person. Author names should not be mentioned.
- By submitting a revision, you attest that you are aware of our payment policies found here: <https://www.life-science-alliance.org/copyright-license-fee>

B. MANUSCRIPT ORGANIZATION AND FORMATTING:

Reviewer #1 (Comments to the Authors (Required)):

1. A short summary of the paper, including description of the advance offered to the field.

Ding et al have developed an epigenetic repressor based on a fusion protein including dead cas9, KRAB module and MeCP2. This work shows effective and persistent repression.

Repression has been reported in mESC, Embryoid bodies, 293T cells. The system was further optimized with the better ZIM3 krab domain. This optimized form was highly successful in mESC and demonstrated moderate activity in primary human T cells.

Curiously, epigenetic repression of this constructs seems not reduced by reducing promoter methylation which invites very interesting followup studies to further elucidate the repression mechanism.

Response: We thank reviewer #1 for the concise summary of our results.

2. For each main point of the paper, please indicate if the data are strongly supportive. If not, explicitly state the additional experiments essential to support the claims made and the timeframe that these would require.

dCas9-KRAB-MeCP2 seems very well supported with FACS measurement and testing in multiple scenarios including different cell types, embryoid bodies, and promoters.

dCas9-ZIM3-MeCP2 superiority to dCas9-KRAB-MeCP2 seems clear from data provided in Fig 3

Repression connection to methylation is solidly studied by the loss of function studies (DNMT3A and B KO; WB and methylation on promoters). Mutants are well characterized (genotype validated, and expression loss) and tests on the mutants produce a clear result.

Response: We appreciate the support for our main points of the manuscript.

3. Lastly, indicate any additional issues you feel should be addressed (text changes, data presentation, statistics etc.).

Regarding figures 1B, 2B, 2C and EV1A all show repression but in different formats ("Fold Changes of GFP negative", "GFP expression (%)", "% of GFP negative", "GFP negative percentage"), perhaps it would be clearer for the reader if the format of the representations gets unified.

Response: We thank the reviewer for this suggestion. Figure 1B illustrates the calculation of GFP repression (fold changes of GFP negative percentage) in time-course experiments shown in Figure 1C and 1D. We believe that for providing a comparable overview of the data, this is the best way to illustrate it. This is why for this figure we would like to keep the format. However, as suggested, we have converted Figure 2B, 2C, 5B and EV1A (now Figure S1A of LSA format) to the unified format as GFP negative %.

Reviewer #2 (Comments to the Authors (Required)):

The current manuscript described a novel Crispr-based method for long term silencing of gene expression in a variety of mammalian cell lines. The authors show

that a dCas9-KRAB-MeCP2 constructs targeted to the promoter region of a transgene or endogenous gene can suppress their expression for long times (up to more than one month). They also show that if target gene expression is driven by a CpG island promoter, silencing occurs alongside with CpG methylation, however also non CpG-promoter genes can be silenced by the same construct; moreover, knocking out DNMT3A and 3B does not avert gene silencing. Overall, this suggests that silencing occurs through a mechanism that does not depend on CpG methylation.

I find this tool extremely useful for the community, as a number of studies would strongly profit from such an amenable approach. Overall the study is well performed and controlled, and results are properly interpreted.

Response: We appreciate the evaluation and comments by reviewer #2.

Given the observed lack of specificity towards CpG island promoters, it would have been nice to test to what extent this cas9 derivative might work as a roadblock for transcription (at least at the beginning of the experiment), for example by utilising different gRNAs targeting more upstream regions of the genes (i.e. not so close to the TSS).

Response: We appreciate the suggestion to investigate the optimal distance of the recruited dCas9-KRAB-MeCP2 by designing a panel of sgRNAs targeting various positions around the TSS. However, we would like to point out that this aspect has been thoroughly investigated by Yeo and colleagues (Yeo et al., 2018). We do not believe that our study would reveal novel aspects in this regard. However, we now point the reader to the Yeo et al. publication on page 4 of our revised manuscript.

Also, the reader is clearly left with the question of how this system works at the mechanistic level. One could think to test histone modifications at the different CpG plus and minus promoters, for example it could be interesting to study H3K9 methylation. Overall, while this study falls short in providing the characterisation of the mechanisms behind dCas9-KRAB-MeCP2-mediated silencing, it seems very likely to provide a powerful new tool to the scientific community.

Response: We are grateful for this suggestion. To address this point, we have performed chromatin IP (ChIP) using an anti-H3K9me3 antibody to determine H3K9me3 levels in Gapdh-GFP in control and sgGFP transfected cells. Indeed, the ChIP-PCR results revealed significant increase of H3K9me3 modification at the dCas9-KRAB-MeCP2 target region, suggesting that long-term gene silencing might be maintained by repressive histone modifications (i.e. H3K9me3). We have integrated this result into revised manuscript as Fig S4.

February 18, 2022

RE: Life Science Alliance Manuscript #LSA-2021-01321-TR

Dr. Frank Buchholz
Centre for Medical Systems Biology
Medical Systems Biology, UCC, Medical Faculty Carl Gustav Carus
Medical Faculty and University Hospital Carl Gustav Carus, TU Dresden
Dresden 01307
Germany

Dear Dr. Buchholz,

Thank you for submitting your revised manuscript entitled "DNA-methylation independent long-term epigenetic silencing with dCRISPR/Cas9 fusion proteins". We would be happy to publish your paper in Life Science Alliance pending final revisions necessary to meet our formatting guidelines.

- please add a Category for your manuscript in our system
- please add the Twitter handle of your host institute/organization as well as your own or/and one of the authors in our system
- please consult our manuscript preparation guidelines <https://www.life-science-alliance.org/manuscript-prep> and make sure your manuscript sections are in the correct order;
- please separate the Results and Discussion section into two - 1. Results 2. Discussion, as per our formatting requirements
- please add your main, supplementary figure, and table legends to the main manuscript text after the references section;
- please add an Author Contributions section to your main manuscript text
- please use the [10 author names, et al.] format in your references (i.e. limit the author names to the first 10)
- please provide a separate Data Availability section

A. FINAL FILES:

B. MANUSCRIPT ORGANIZATION AND FORMATTING:

Sincerely,

February 28, 2022

RE: Life Science Alliance Manuscript #LSA-2021-01321-TRR

Dr. Frank Buchholz
Centre for Medical Systems Biology
Medical Systems Biology, UCC, Medical Faculty Carl Gustav Carus
Medical Faculty and University Hospital Carl Gustav Carus, TU Dresden
Dresden 01307
Germany

Dear Dr. Buchholz,

Thank you for submitting your Research Article entitled "DNA-methylation independent long-term epigenetic silencing with dCRISPR/Cas9 fusion proteins". It is a pleasure to let you know that your manuscript is now accepted for publication in Life Science Alliance. Congratulations on this interesting work.

DISTRIBUTION OF MATERIALS:

Again, congratulations on a very nice paper. I hope you found the review process to be constructive and are pleased with how the manuscript was handled editorially. We look forward to future exciting submissions from your lab.

Sincerely,
